# Influence of Temperature on Exciton Dynamic Processes in CuPc/C60 Based Solar Cells

**DOI:** 10.3390/mi12111295

**Published:** 2021-10-22

**Authors:** Lijia Chen, Lun Cai, Lianbin Niu, Pan Guo, Qunliang Song

**Affiliations:** 1College of Physics and Electronics Engineering, Chongqing Normal University, Chongqing 401331, China; ljchen01@cqnu.edu.cn (L.C.); niulb03@126.com (L.N.); guopan@cqnu.edu.cn (P.G.); 2School of Materials, Sun Yat-sen University, Shenzhen 518107, China; cailun@mail2.sysu.edu.cn; 3Institute for Clean Energy and Advanced Materials, School of Materials and Energy, Southwest University, Chongqing 400715, China

**Keywords:** copper phthalocyanine, low temperature, absorption, exciton, organic solar cell

## Abstract

Although the effect of high temperature on the performance of organic solar cells has been widely investigated, it is inevitably influenced by the associated annealing effect (which usually leads to film morphology change and variation in electrical properties), which makes the discussion more sophisticated. In this study, we simplified the issue and investigated the influence of low temperatures (from room temperature to 77 K) on the photocurrent and internal/external quantum efficiency of a CuPc/C60 based solar cell. We found that besides the charge dynamic process (charge transport), one or more of the exciton dynamic processes, such as exciton diffusion and exciton dissociation, also play a significant role in affecting the photocurrent of organic solar cells at different temperatures. Additionally, the results showed that the temperature had negligible influence on the absorption of the CuPc film as well as the exciton generation process, but obviously influenced the other two exciton dynamic processes (exciton diffusion and exciton dissociation).

## 1. Introduction

The exciton and charge dynamic processes, including (i) exciton generation, (ii) exciton diffusion, (iii) exciton dissociation, (iv) free charge transportation and (v) charge collection are five important processes that determine the performance of organic solar cells (OSCs) and perovskite solar cells (PSCs) [1,2,3]. Though the power conversion efficiency (PCE) of OSCs has been found to be over 17% [4], further improvement in performance has encountered a bottleneck due to the limited understanding towards how these dynamic processes determine the performance of OSCs. To explore this issue, some experimental methods have been developed, such as inserting a buffer layer [5,6], tailoring the thin film morphology [7,8], optimizing the active layer deposition [9], or designing annealing methods [10]. Among these methods, temperature control has been regarded as one of the most important approaches thanks to its significant impact on the performance of OSCs [11,12]. For example, employing proper thermal annealing can increase the PCE of OSCs by more than 50% [13]. Generally, raising the device or fresh coated substrate temperature (usually 60–250 °C) promotes the molecule stacking, and enhances the thin film crystallinity, which leads to higher carrier mobility and reduced series resistance at the same time [14,15]. According to the conventional understanding, the increased short circuit current (Jsc) at higher temperature has been mainly ascribed to the increase in charge mobility after the thermal annealing treatment [13]. Meanwhile, in addition to the study on the mentioned high temperature effect, some other groups investigated the low temperature effect. They found that the short circuit current decreases when lowering the temperature [16]. Similarly, they ascribed the low temperature effect to the decrease in charge mobility caused by the temperature decrease. These reported explanations can be summarized as the effect of temperature on performance of OSCs via the influence of temperature on the charge dynamic process [13,16]. However, it is worth pointing out that before generating free charges, there are three exciton dynamic processes, namely, exciton generation, exciton diffusion and exciton dissociation [17]. Although the exciton dynamic processes are important in determining the performance of OSC, quite limited theoretical and experimental work was reported in studying how temperature influences exciton dynamic processes [18,19]. Studies related to the optical feature investigation revealed that for some specific organic thin films, such as pentacene and poly (3-hexylthiophene) (P3HT), their absorptions are closely related to temperature [20,21], demonstrating that the efficiency of the exciton generation could be improved by selecting proper temperature.

Some studies related to the optical feature investigation revealed that for some specific organic thin films, such as pentacene, their absorptions are closely related to temperature, demonstrating that the efficiency of the exciton generation could be improved by selecting the proper temperature [20,22]. Furthermore, the exciton diffusion and exciton dissociation processes were qualitatively depicted to be inefficient at low temperature for some given OSCs [18,19,23]. Exciton diffusion length dropped from 4.5 to 3.2 nm in the conjugated polymers upon cooling the sample from room temperature to 150 K [24].

In this work, we found that the dependence of the exciton generation with temperature was different in CuPc thin film. Unlike that in pentacene thin film, the influence of temperature on the absorption of CuPc is almost negligible, demonstrating that exciton generation and charge collection are not affected. The study on the influence of temperature on the exciton dynamic processes is more complicated than that on the charge dynamic processes. Meanwhile, external quantum efficiency (EQE) of CuPc/C60 based OSCs monotonically decreased with a decreasing temperature. For example, compared to its initial value at room temperature, the EQE at 77 K was reduced by 34%, revealing that the significant reduction of photocurrent caused by lowering temperature in CuPc/C60 based OSCs is mainly ascribed to the inefficient exciton diffusion and dissociation.

## 2. Experimental

All organic materials were purchased from Sigma-Aldrich Co. and used as received without further purification. The OSCs were fabricated on the pre-cleaned glass substrates coated with patterned indium tin oxide (ITO) with the square resistance of ~15 Ω/sq. The cleaned substrates were then oxygen plasma treated for ~5 min to remove all organic residuals and increase the work function. The structure of the device is ITO/CuPc (20 nm)/C60 (55 nm)/tris-8-hydroxy-quinolinato aluminum (Alq3) (6 nm)/Al (100 nm). CuPc, C60 and Alq3 were thermally deposited onto the ITO anode successively in an ultra-high vacuum chamber with a base pressure of 2 × 10^−6^ Pa. The Al strip was then deposited through a shadow mask in the same chamber as the cathode. The active area was 9 mm^2^. The 50 nm CuPc film was deposited on quartz substrates for optical absorption measurements. A tungsten lamp without/with a 600 nm long-wave pass optical filter (LP600) provides the full band/given band (with the wavelengths over 600 nm) light to illuminate the fabricated samples. The J–V characteristics, under illumination of lights with different wavelengths, were measured by using the Keithley 2400 system in a vacuum (better than 1 × 10^−4^ Pa) probe station (TTPX, LakeShore) chamber. The light intensity from the tungsten lamp was checked by a calibrated silicon detector after each J-V measurement, showing negligible intensity variation (less than 2.8%) during the whole experiment. The probe station provides accurate temperature control of 0.1 K and high temperature uniformity (maximum 0.5 K variation, even the temperature was lowered to 77 K). The EQE was calculated from the photocurrent generated from the monochromatic light measured by a lock-in amplifier (SR830). Monochromic light was generated by a xenon light coupled with a monochromator and modulated by a mechanical chopper. The absorption of CuPc thin film at low temperatures was calculated from log (I_0_/I) by using the determined light intensity before (I_0_) and after (I) the film. The light intensity was measured by a calibrated silicon photodetector. The absorption of the device containing the Al electrode at room temperature was conducted using the relative reflectance accessory of an UV–Visible spectrophotometer (UV-2550). An ITO coated glass with 100 nm Al was used as the reference.

## 3. Results and Discussion

The structure of CuPc/C60 based OSCs and its corresponding energy level are shown in Figure 1. The performance of the device was first evaluated under 100 mW/cm^2^.

AM1.5G simulated sunlight at room temperature, which was followed by transfer to the probe station immediately for further measurements. To make sure thermal balance was reached between the device and sample stage, measurements at each temperature were also carried out 24 min later, after the sample stage reached a steady state. Measurements were conducted for temperatures ranging from 77 up to 300 K and then in the reverse order, that is from 300 K down to 77 K. To the best of our knowledge, the electrical performance of OSCs has not been investigated at this low temperature range. No significant differences in current readings were found over the investigated temperature range. In addition, the degradation of the device’s performance was negligible during the measurements; the J-V characteristics under AM1.5G, 100 mW/cm^2^ at room temperature, after measurements in the probe station, displayed a short circuit current, Jsc, of 4.09 ± 0.07 mA/cm^2^, an open circuit voltage, Voc, of 0.49 ± 0.02 V, and a fill factor, FF, of 57 ± 2%, which was the same performance as before loading the device in the probe station.

Figure 2a,b show the J-V characteristics of the device at different temperatures, under direct illumination of a tungsten lamp or along with a 600 nm long-wave pass optical filter (LP600), which attenuates wavelengths shorter than 600 nm. It is worth pointing out that the photocurrent obtained with the LP600 filter comes from CuPc film, since C60 has negligible absorption beyond 600 nm [25].

As shown in both Figure 2a,b, the Voc decreases while the Jsc increases with the increasing temperature, and this is consistent with reported works [26]. Note that, Voc at 77 K is about 0.67 V, which is very close to the energy difference (0.7 eV) between the highest occupied molecular orbital (HOMO) of CuPc (5.2 eV) and the lowest unoccupied molecular orbital (LUMO) of C60 (4.5 eV) [27]. However, this is obviously larger than the work-function difference (0.52 eV) between the used electrodes (oxygen plasma treated ITO is around 4.8 eV while Al is 4.28 eV) [28]. The above observation confirmed that Voc is determined by the energy difference between donor and acceptor rather than the work function difference of the two electrodes [16,29].

To get deeper insight into the influence of temperature in the device, the EQE spectrum at different temperatures was recorded. As displayed in Figure 3a, the EQE value monotonically increases when the temperature increases from 77 to 300 K, and this is consistent with the results shown in Figure 2. The EQE values in the spectral range 550–750 nm typically reveal the contribution of CuPc to the photocurrent. By choosing the EQE at 77 K as the reference, we can obtain the relative variation of EQE for each temperature. As shown in Figure 3b, a maximum increase of 34% in EQE is achieved at 645 nm when the temperature increases from 77 to 300 K. The relative increase of EQE with temperature at different wavelengths is shown in Appendix A.

As mentioned before, the measured *J_sc_* and EQE are typically determined by processes of exciton generation, diffusion and dissociation as well as free charge transportation/collection. To reveal the contribution of each process to the photocurrent, the EQE is given by the following formula [21]:(1)$$\mathrm{EQE}=\eta_{A}\eta_{ED}\eta_{CT\;}\eta_{CC}$$
where $\eta_{A},\;\eta_{ED},\eta_{CT},\;and\;\eta_{CC}$ are the exciton formation efficiency of the photoactive region, the percentage of photo-generated excitons that diffuse to the donor-acceptor (D-A) interface, the percentage of exciton dissociation at the D-A interface and the efficiency of free charge transportation/collection, respectively.

To investigate the influence of temperature on $\eta_{A}$, the absorption spectra of CuPc film (~50 nm) and the relative changes of absorption efficeicy have been recorded at different temperatures, which is shown in Appendix A. As reported in Figure 4a, the absorption variation is negligible in the range of 77–300 K. To the best of our knowledge, the temperature dependence of the absorption spectra of CuPc film has not been reported previously. Appendix A shows the maximum relative absorbance (with the absorbance at 77 K chosen as reference) change at 300 K is less than 2%, and this occurs at 620 nm and 700 nm, which are the peak absorption wavelengths of CuPc. A simple calculation shows that 2% change of absorbance induces less than 3% change in $\eta_{A}$.

Figure 4b shows the absorption spectra and $\eta_{A}$ of the OSCs device (structure: ITO/CuPc (20 nm)/C60 (55 nm)/Alq_3_ (6 nm)/Al) at room temperature using an UV-Vis spectrophotometer with a relative reflection measurement option. The absorbance of the OSC device at the peak absorption wavelengths, 620 nm and 700 nm, is about 1.6 times higher than that of the 50 nm CuPc film on quartz, which is due to the photon re-absorption caused by the strong reflection of the Al electrode of the OSC device [30,31]. It is reasonable to assume that the temperature dependence of the absorption, and hence $\eta_{A}$, of the device in the range of 550–750 nm is the same as that of the thin film shown in Figure 4a. Hence, the relative variation of absorbance as well as $\eta_{A}$ of the OSCs device is negligible compared to the change in EQE attained over the same temperature range. Therefore, the absorption efficiency ($\eta_{A}$) variation, and hence the exciton generation change, is not the main reason for the observed increase in EQE with increasing temperature displayed in Figure 3.

We will now shift our discussion to the influence of temperature on *η_CC_*. It has been reported that the mobility of organic thin films like CuPc can increase several orders of magnitude when increasing the temperature from ~100 K up to 300 K [32,33,34]. If the efficiency of the free charge transportation/collection is proportional to the carrier mobility, and if we assume that this is the only reason for the EQE increase, then the EQE will saturate at certain temperature, at which all carriers are collected. Contrary to this assumption, an everlasting continuous increase of EQE was observed when the temperature was increased from 77 to 300 K, as shown in Figure 3. Therefore, it is reasonable to assume that all free carriers were fully collected by the electrodes even at 77 K. In other words, $\eta_{CC}$ is ~100% and it is not sensitive to temperature. Such a result is in agreement with other reports that the collection efficiency is ~100% due to the built-in electric field [30].

The internal quantum efficiency (IQE) of the device, calculated using the expression IQE = EQE/absorptance [35], is shown in Figure 5a for different temperatures. The IQE for the OSC device is below 25%, revealing that most of the excitons have no contribution to the final photocurrent. Note that, although the CuPc/C60 based planar OSC device was not optimized for the best performance, it does represent a typical performance for this OSC type. There are two possible reasons for this low quantum efficiency, namely, the inefficient exciton diffusion to the D-A interface and the low efficiency of exciton dissociation at the D-A interface.

The exciton diffusion length in conjugated polymers was reported to increase from ~3 nm to ~4.5 nm when the temperature increases from 4 to 293 K [36]. It has also been demonstrated that higher temperature facilitates exciton dissociation [19]. The low IQE of the device indicates that there is only a fraction of excitons available for diffusing to and/or dissociating at the D-A interface. Thus, the exciton diffusion efficiency ($\eta_{ED}$) and/or the dissociation efficiency ($\eta_{CT\;}$) are believed to be the main contributions to the EQE changes at different temperatures for the CuPc/C60 device. However, we cannot further separate the contribution of exciton diffusion and exciton dissociation to the photocurrent variation because it is difficult to obtain the exciton diffusion length and exciton dissociation efficiency at different temperatures.

Figure 5b schematically shows the relative variation of the η parameters with different temperatures when parameters at 77 K were chosen as reference. Its physical interpretation is described as follows: the amount of photo-generated excitons in CuPc film is maintained at almost the same level when the temperature increases from 77 to 300 K, while exciton diffusion and/or dissociation increase with temperature. Excitons diffuse to the CuPc-C60 interface and then are finally dissociated into free carriers. The free charge transportation and collection efficiency is nearly 100%, even at 77 K, for the CuPc/C60 device. The photocurrent increase, at higher temperature, in the CuPc/C60 device is thus the result of enhanced exciton diffusion and/or dissociation, which results in more free carriers.

## 4. Conclusions

The J-V and EQE characteristics of CuPc/C60 organic solar cells were studied over a low temperature range (77–300 K). Experimental results showed that the photocurrent monotonically increased with temperature. The absorption of a 50 nm evaporated CuPc thin film at low temperatures was investigated for the first time, which showed negligible absorbance change in the range of 77–300 K. The observed increase over 25% in EQE when the temperature increases from 77 to 300 K was attributed to better exciton diffusion and/or dissociation rather than only higher mobility for better carrier transportation and collection at high temperatures. Further investigation is expected to quantitatively clarify the contributions of exciton diffusion and exciton dissociation to the photocurrent variation at different temperatures, and implement to other solar cells, such as perovskite solar cells.

## Figures and Tables

**Figure 1 micromachines-12-01295-f001:**
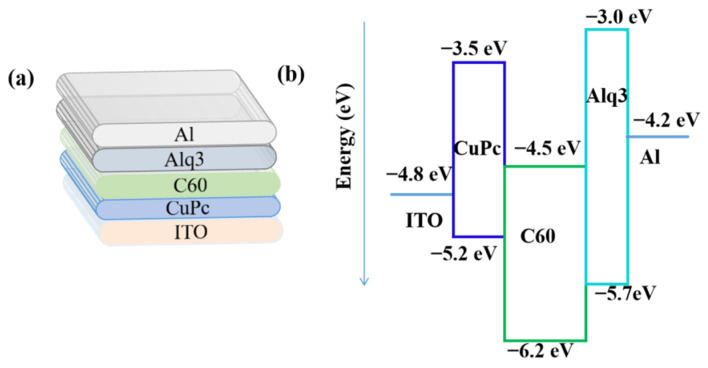
The device structure (**a**) and energy level (**b**) with ITO/CuPc (20 nm)/C60 (55 nm)/Alq3 (6 nm)/Al (100 nm).

**Figure 2 micromachines-12-01295-f002:**
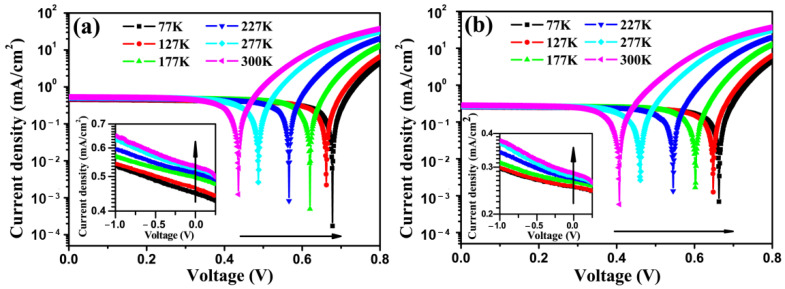
J-V characteristics of OSCs at various temperatures, under illumination of tungsten lamp light (**a**) without and (**b**) with a 600 nm long-wave pass optical filter. The insets show details of J-V curves in the range of −1~0.25 V. Device structure is: ITO/CuPc (20 nm)/C60 (55 nm)/Alq3 (6 nm)/Al (100 nm).

**Figure 3 micromachines-12-01295-f003:**
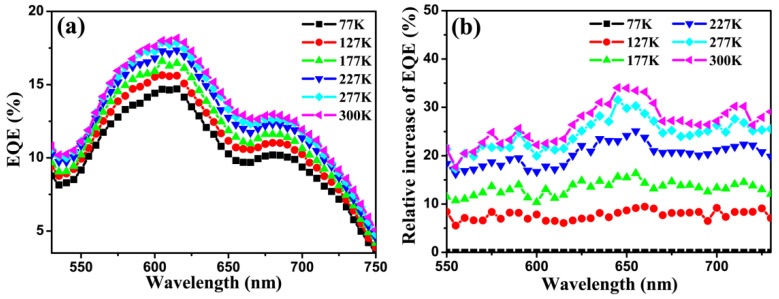
(**a**) EQE of the developed device at various temperatures. (**b**) Relative variation of EQE for different temperatures with the EQE at 77 K being chosen as reference.

**Figure 4 micromachines-12-01295-f004:**
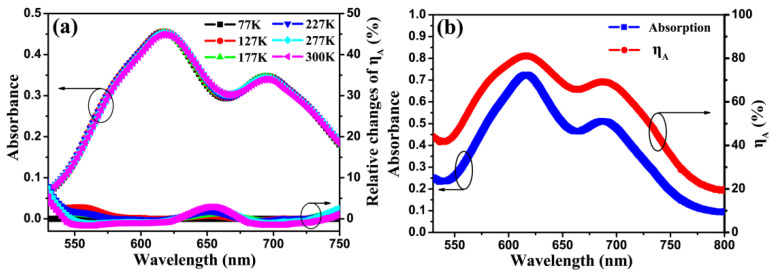
(**a**) Absorption spectra for a 50 nm CuPc thin film thermal evaporated on quartz substrate and the relative changes of absorption efficiency ($\eta_{A})$ at different temperatures. (**b**) The absorption spectra and $\eta_{A}$ of the developed OSC device at 300 K.

**Figure 5 micromachines-12-01295-f005:**
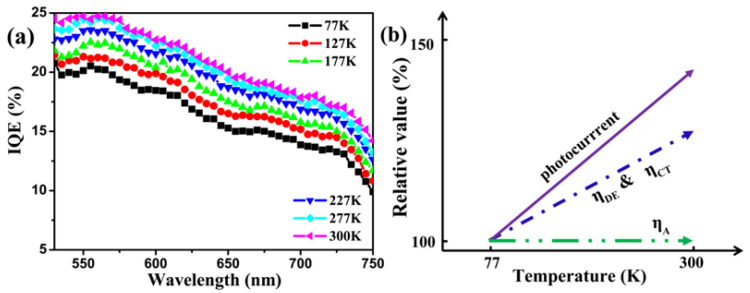
(**a**) Calculated IQE spectra of the developed device at different temperatures; (**b**) Schematic drawing showing absorption efficiency ($\eta_{A})$, exciton diffusion ($\eta_{ED}$) and dissociation efficiency ($\eta_{CT}$), carrier mobility and photocurrent trends versus temperature. 77 K was chosen as the reference temperature.

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
