# Peer review of "Influence of Temperature on Exciton Dynamic Processes in CuPc/C60 Based Solar Cells"

_micromachines, 2021, doi:10.3390/mi12111295_

Round 1

Reviewer 1 Report

The authors investigate the influence of low temperatures (from room temperature to 77 K) on the photocurrent and internal/external quantum efficiency of copper phthalocyanine (CuPc)/ fullerene (C60) based solar cells. This work is interesting and well presented. I would recommend publication of this work in Micromachines after addressing the following issues:

  1. The title of this work using “a CuPc/C60 based solar cell” is not suitable. I suggest the authors to revise the “a CuPc/C60 based solar cell” to “CuPc/C60 based solar cells”
  2. The authors are suggested to cite some references on solar cells published by Micromachines or some recent works on solar cells such as Adv. Energy Mater., 2021, 2101045 and Journal of Semiconductors 2021, 42 (6), 060202.
  3. Fig. 5: the carrier mobility curve is not explained. It is an unrelated graph for this work. The authors could delete it.

Reviewer 2 Report

Manuscript is recommended for publication. 

Reviewer 3 Report

This article is very short and does not content enough relevant new science to my opinion.

The described system is a very old one, not in scope of today research and giving low PCE.

To be able to publish these results, I suggest to the authors to compare them to several other systems they can study. Furthermore, this version seems to be a working version as some non-usefully lines are remaining (lines 71-87, lines 116-120, line 136).

Reviewer 4 Report

This work studies the temperature dependence of organic solar cell performance. The authors have found that the absorption process has negligible change when temp going from 77K to 300K. On the contrary, the temp influenced the exciton diffusion and exciton dissociation process, which results in the difference device performance at different temp. Overall, this work has in-depth physical analysis and good design of experiment, should be accepted with minor revision.

  1. The authors concluded that the temp influenced the exciton diffusion and dissociation process. However, no experimental data provided to prove. Three essential/important experiments should be done and provided in the revised manuscript, temp-dependent exciton diffusion length, dissociation efficiency and electron and hole mobility.
  2. As the temp increases, the photo-current increases which is due to both the exciton process change as well as the free carrier mobility increases. Could the author distinguish the influence of these two parameters and explain which one is more important when solar cell operation temp is changing?
  3. Line71-87 (Materials and Methods) is not relevant to this study, which should be removed or replaced.

Round 2

Reviewer 3 Report

Accept in present form